# Genetic diversity and relationship of Bugesera and Rwamagana indigenous chicken populations with SASSO chickens using DArTseq SNPs

Valentin Mujyambere[1]*, Kwaku Adomako[2], Martin Ntawubizi[1], Laetitia Nyinawamwiza[1], Judith Uwihirwe[1], Alexander Wireko Kena[2]

1 Department of Animal Production, University of Rwanda, Nyagatare, Rwanda, 2 Department of Animal Science, Kwame Nkrumah University of Science and Technology, Kumasi, Ghana

* mujyasval@gmail.com

## Abstract

The genetic improvement of Rwandan indigenous chickens (IC) is becoming a priority and requires genetic characterization as the foundation for genetic improvement. The aim was to study the genetic diversity and relationship of IC from Bugesera and Rwamagana districts with SASSO chickens in Rwanda. Eighty-seven DNA samples were genotyped using Diversity Array Technology and generated 45,677 DArTseq SNPs. After the quality control, the genetic diversity and relationship were analyzed with dartR package, STRUCTURE and STRUCTURE Harvester. Only 8087 markers and 86 samples were remained for analysis. The high and low expected heterozygosity (He) were observed in Rwamagana (0.405) and SASSO (0.388) populations, respectively. The lowest difference was observed between SASSO and Rwamagana populations (0.018). The structure analysis separated IC (Cluster 1) from SASSO chickens (Cluster 2). Chickens from Rwamagana population were genetically related to SASSO indicating the uncontrolled crossbreeding. DArTseq SNPs were efficient and can be recommended for genomic studies in chickens.

## Introduction

In 2017, the contribution of the agricultural sector in Rwanda, including livestock, was estimated at approximately 31% of the country's gross domestic product (GDP), as reported in the Poultry Sector Analysis of Rwanda (S1 File). In the same year, according to the Rwanda Livestock Master Plan (LMP), developed in 2017 by the International Livestock Research Institute (ILRI) in collaboration with the Ministry of Agriculture and Animal Resources (MINAGRI) and the Rwanda Agriculture and Animal Resources Development Board (RAB), the livestock subsector accounted for approximately 7 million chickens (S2 File). There was an increase in the poultry population, with 5.3 million chickens as reported in the same document (S2 File) and by National Institute of Statistics of Rwanda (S3 File), 70% of those birds being

**Data availability statement:** All relevant data are within the paper and its Supporting information files.

**Funding:** The author(s) received no specific funding for this work.

**Competing interests:** The authors have declared that no competing interest exist.

indigenous chickens (ICs). Mbuza et al. [1] reported that approximately 92.4% of households interviewed keep indigenous chickens, whereas 7.6% keep exotic or improved strains. The flock size at household level was estimated to be between 2 and 20 birds, but the size of less than 10 birds per rural household was predominant [1,2]. National Institute of Statistics of Rwanda (S4 File) reported the total production of 117,430,896 eggs in 2020, while it was estimated that 8,272 tons of total eggs and 38,845 tons of total chicken meat were produced in 2020 as reported in the Poultry Sector Analysis of Rwanda, cited above. Although ICs constitute the majority of poultry produced in Rwanda, their production is too low; however, it accounts for 32% and 34% of the eggs and chicken meat produced, respectively, as reported in LMP. Indigenous chickens are predominantly managed in an extensive system, and the breeding practices under this system are very complicated [3]. Most of the IC chickens kept by rural households for flock replacement are bought from village markets, neighbors or from their own flocks [1,4]. The consequence of this breeding system is a decrease in genetic variability in favor of a high level of inbreeding. The LMP, mentioned above highlighted a strategy to supply good breeding stock to farmers. Two subsystems have been proposed to improve the family chicken production system, namely, improved traditional family chickens (ITFCs) and crossbred family chickens (CFCs). The aim of LMP was to reduce red meat consumption in favor of chicken and pork consumption. The LMP expected that the production of eggs would increase to 1.9 billion in 2032, with 5.7 kg or 114 eggs per capita. Improving the traditional family system of increasing ICs will contribute greatly to meeting these expectations. The two production systems proposed by the LMP, are expected to be used for IC genetic improvement: selection from the existing IC and crossbreeding between IC and exotic breeds. However, the ICs must be well morphologically and genetically characterized for better valorization of their genetic variability in breeding and conservation programs. Two studies on morphological characterization [4,5] and three studies on genetic diversity and population structure have been published. The first study on genetic diversity used microsatellite markers [6], the second used single nucleotide polymorphism (SNP) markers [7], and the third used Diversity Array Technology (DArT) [8]. Rwandan IC were classified into four gene pools based on genetic similarity [6]. This study demonstrated that IC from the Eastern of the country were clustered alone, suggesting the large population size and the low gene inflow. The IC in North West and Central North grouped together, highlighting the high gene flow between two agro-ecological zones, especially through the exchange of reproductives. The similar trend was observed in IC from South West and Central South which clustered together. The fourth gene pool comprised IC from South West in admixture with exotic chickens suggesting the occurrence introduction of exotic breeds in the area aiming at the genetic improvement of IC through crossbreeding. The study conducted in Eastern province of Rwanda by Mujyambere et al. [8] detected subpopulations among IC, reinforcing the existence of unique gene pools even within one region. The high expected heterozygosity (0.658) observed in the study by Habimana et al. [6], and the number of alternate allele counts (21 alleles per locus) observed in the study by Mujyambere et al. [8] suggested a high genetic variability

within IC populations. The low inbreeding coefficient of 0.04 observed within IC [6] and the absence of extreme divergence (0.010–0.056) among IC populations [8] indicated the absence of inbreeding bottleneck and the broadness of the genetic diversity. Findings from these studies were relevant to LMP objectives by demonstrating the existence of high genetic variability among IC populations confirming that the genetic improvement can occur within IC populations without reducing the diversity although distinct gene pools for breeding programs focusing on localized-particular traits were defined. In addition to genetic variability studied, the quantitative trait loci (QTL) associated with body weight of IC and their immune response to Newcastle disease were identified [7] to assist in breeding selection of reproductives resisting to this viral disease. However, for enhancing the IC conservation, supply of characterized IC to rural household farmers, and promote the crossbreeding between IC and exotic breeds for genetic improvement, four IC breeds were recently morphologically and genetically characterized, namely: Inshenzi, Inganda (Dwarf), Umurangi (Naked neck), and Indayi [9]. Recently, exotic breeds have been introduced in Rwanda, particularly in Eastern Province. The recent and spread breed was SASSO. These SASSO chickens, along with other exotic breeds, were kept together with indigenous chickens so that they crossed without control [3]. Nevertheless, the implementation of the two breeding subsystems described above requires the use of pure breeds, which require genetic characterization. Therefore, further studies are needed to confirm the existence of genetic variability in ICs and their relationships with a range of exotic breeds already available across the country. This study used DArTseq SNP markers to study the genetic diversity and population structure of IC in the Eastern Province of Rwanda. DArTseq SNPs, similar to other DArT markers, such as SilicoDArTs, were discovered and sequenced in the whole genome of plants [10,11] and were found to be biallelic and codominant markers scored as present [1] or absent (0) [12–14]. DArTseq SNPs are suitable for genetic diversity and population structure studies because of their numerous advantages, such as high-throughput genotyping, cost-effectiveness, no prior sequence information needed, simplicity and speed, accuracy in population studies, and low DNA input [12,15]. DArT markers have been shown to be effective for genomic studies in chickens [8]. The aim was to study the genetic diversity and relationships of indigenous chickens in Bugesera and Rwamagana populations with SASSO chickens via DArTseq SNPs in Rwanda. Understanding the relationship between indigenous chickens from two populations and SASSO chickens will provide insight into the effectiveness of breeding methods and aid in decision-making before planning breeding and conservation programs.

## Materials and methods

### Study area

The study area covered the Bugesera and Rwamagana distrcits of the Eastern Province of Rwanda [8]. Both districts border the City of Kigali. Bugesera is located between 30° 05' East Longitude, and 2° 09' South Latitude, whereas Rwamagana lies between 30° 26' East Longitude and 1° 57' 9" South Latitude. The ambient temperature ranged from 26°C to 29°C and from 19°C to 30°C for Bugesera and Rwamagana, respectively.

### Chicken blood sampling procedure

Eighty-seven (87) chickens were selected for blood sample collection. The selected chickens originated from three populations, the Bugesera and Rwamagana populations, from the Bugesera and Rwamagana districts of the Eastern Province of Rwanda, and the SASSO population. The sample for blood collection was composed of 30 birds from Bugesera, 47 birds from Rwamagana, and 10 birds from SASSO. As mentioned in the study of [8], the SASSO population was composed of 10 SASSO chickens. The birds were collected from different rural households, with one bird from each at a distance of more than 500 m, to avoid the sampling of related birds with reference to the sampling protocol proposed by [16]. SASSO chickens were selected from two flocks of private farmers, one flock from each district. Three reasons explain the choice of these populations. The first reason was the contribution to poultry value chain development in the Eastern province of Rwanda, one of the objectives of the SEAD (Strengthening Education for Agricultural Development) project

before it closed in June 2021. The second reason was based on the location of two districts bordering the City of Kigali as potential markets for poultry products. The third reason was the increase in gene flow due to the spread of SASSO chickens among rural farmers by Uzima chicken Ltd. Indigenous chickens of both sexes (males and females) were sampled at ages ranging from 8 to 10 months, whereas only SASSO females were sampled at 9 months of age. ICs were managed at households in rural villages, and their mating was random but not controlled. SASSO purebred chickens were maintained in an intensive system at private egg production farms where they were fed layer feed. Blood samples of 2 ml each were collected from the wing vein in a 4 ml-EDTA tube by a licensed and professional veterinarian with 21Gx1 1/2 (MED-833) multi-drawing needles after disinfecting the sampling site with a cotton imbibed alcohol. A vaccine carrier box was used to carry tubes containing blood samples for transport at ambient temperature. Before any treatment, the samples were kept in a refrigerator in the laboratory of Biology of the University of Rwanda, College of Science and Technology (CST), at 4°C. Within less than 24 hours, with the use of pipettes and pipette tips, under sterilization conditions, blood samples of 1μl were collected, with one sample from each EDTA tube being dropped into 87 wells of a 96-well PCR plate. After being covered and sealed, the 96- well PCR plate containing the blood samples was transported at ambient temperature and sent to SEQART Africa located at Biosciences Eastern and Central Africa at the International Livestock Research Institute (BecA-ILRI) Hub, formerly the Integrated Genotyping Service and Support (IGSS) platform, in Nairobi Kenya for genotyping. The guidelines of the revised Animals (Scientific Procedures) Act 1986 were followed for all chicken handling and for participation of bird owners, and all methods were implemented in accordance with the relevant guidelines and regulations of the Declaration of Helsinki. Consequently, before the study began, the research protocol was approved by the Research Screening and Ethical Clearance Committee of the College of Agriculture, Animal Sciences and Veterinary Medicine, University of Rwanda, and the issued Research Ethical Clearance reference number was 033/19/DRI. The study was reported in accordance with ARRIVE guidelines (https://arriveguidelines.org).

### Informed consent statement

Informed consent was obtained from all bird owners involved in the study. The title of the study, simple and short introduction and purpose, voluntary participation, procedures (blood samples collection), some risks and discomforts the procedures may cause to the chickens, benefits and confidentiality were explained to participants and included into the informed consent documents.

### Inclusivity in global research

Additional information regarding the ethical, cultural, and scientific considerations specific to inclusivity in global research is included in the Supporting Information (S1 Checklist).

### DNA extraction and genotyping by sequencing

In the laboratory, as described in the previous study of [8], the blood samples were subjected to the DNA extraction process. Nucleomag is the method used to extract DNA from blood samples, where 50−100 ng/μl is the range of the amount of extracted genomic DNA. Agarose (0.8% agarose) was used to check the quantity and quality of the extracted DNA. PCR amplification of adapter-ligated fragments following the DArTSeq complexity method through digestion of genomic DNA and ligation of barcoded adapters was used to construct libraries according to the protocol detailed by [12]. Single-read sequencing runs for 77 bases were used to sequence libraries already constructed, and Hiseq2500 was used in the next-generation sequencing procedure. This sequencing process uses genotyping by sequencing (GBG), a DArTseq technique that provides, even from most complex polyploid genomes, high-quality, rapid and affordable genome profiling, as reported previously [12]. DArTsoft14, an in-house marker scoring algorithm-based pipeline, was used for DArTseq marker scoring. SNP markers are a type of DArTseq marker binary scored for the presence/absence (1 and 0,

respectively) of the restriction fragment in the genomic representation of the sample. This DArTseq marker system was aligned with the reference genome of chicken v5 to identify chromosomes and positions (Gallus_gallus-5.0 - Genome - Assembly - NCBI (nih.gov).

**Genetic diversity and population structure**

Missing data are sometimes challenging in statistical data analysis. For this reason, the DArT report generated by the genotyping process (S1 Table) was imputed for missing data KDCompute online software beta 1.5.2. (https://kdcompute.seqart.net/kdcompute/login). It was launched by Diversity Arrays Technology Ltd, in 2017. The imputation was performed with simple matching coefficient (SMC) of 0.743 generated via the null method. The method was run on a dataset with an additional 10% introduced missing values. The imputed, introduced missing values were then compared to the original dataset to calculate a Simple Matching Coefficient (SMC). DArTseq SNP markers were checked for their quality before any analysis performed via the filter option of MS Excel. Markers without identified positions and sex chromosome-linked markers were removed before data analysis. Markers with a call rate < 95%, one ratio < 5%, polymorphic information content (PIC) for both the reference and SNPs < 0.1, and RepAvg (average reproducibility) < 90% were also discarded from the data before analysis. As explained in S1 Table, the call rate is described as the proportion of samples for which the genotype call is either "1" or "0". One Ration indicates the proportion of samples for which the genotype score is "1". RepAvg or average reproducibility represents the average proportion of technical replicate assay pairs for which the marker score is consistent. Individuals with a call rate < 0.90 were also removed from the data before analysis. Microsoft Excel was used for call rate and PIC calculations and chart presentation. The dartR.2.0.4. package of R software [17] was used to count the alleles and the distribution of the alternative SNP alleles within populations, and within SASSO chickens, whereas KDCompute software was used for mapping loci on the first 28 chromosomes. The genetic diversity and relationships among populations were estimated via Nei's 1972 genetic distance [18] and Fst generated via the dartR package and GenAlex.6.503 [19,20], which were used to calculate the expected heterozygosity (He), observed hetero-zygosity (Ho) and fixation index (F). However, the "StAMPP" package [21] and "cluster" package [22] of R software were used to generate Nei's 1972 distance matrix and hierarchical dendrograms, respectively. Principal Component Analysis (PCA) was performed via the dartR package, which was also used in dependency of the ade4 package [20] of R software for Analysis of Molecular Variance (AMOVA). The Bayesian clustering method, which assumes independence between loci and the admixture model, was used to determine the population structure of the chicken genotypes via STRUCTURE 2.3.4 software [23]. For the population structure, after a burn-in period of 50,000 iterations of each cluster, with k values varying from 1 to 6, four runs of 50,000 Markov Chain Monte Carlo (MCMC) iterations were executed. Evanno's Δk method of STRUCTURE Harvester vA.2 [24,25] was used to determine the real number of clusters, symbolized by k.

## Results

**DArTseq SNP marker quality and distribution in the chicken genome**

In total, before the quality screening process, 45,677 DArTseq SNP markers were generated (S1 and S2 Tables). After quality control of the DArTseq SNP markers based on the removal of sex chromosome-linked loci and loci without deter-mined positions, call rates, PICs, RepAvg, and genotypes with call rates < 0.90, 8087 DArTseq SNP markers and 86 chicken genotypes remained for further analysis. Before proceeding to the analysis, the quality was checked before and after the screening (Fig 1). The average call rate (Fig 1a) improved as it increased from 0.89 to 0.98. This increase in call rate was a result of the improvement in the call rate at all levels after quality control. The same improvement was observed in the polymorphic information content (Fig 1b).

A large improvement in the PIC values ranged from 0.4 to 0.5, with an increase from 15% of the markers before screening to 78% of the markers after quality control, and approximately 95% of the DArTseq loci presented a PIC ranging from 0.3

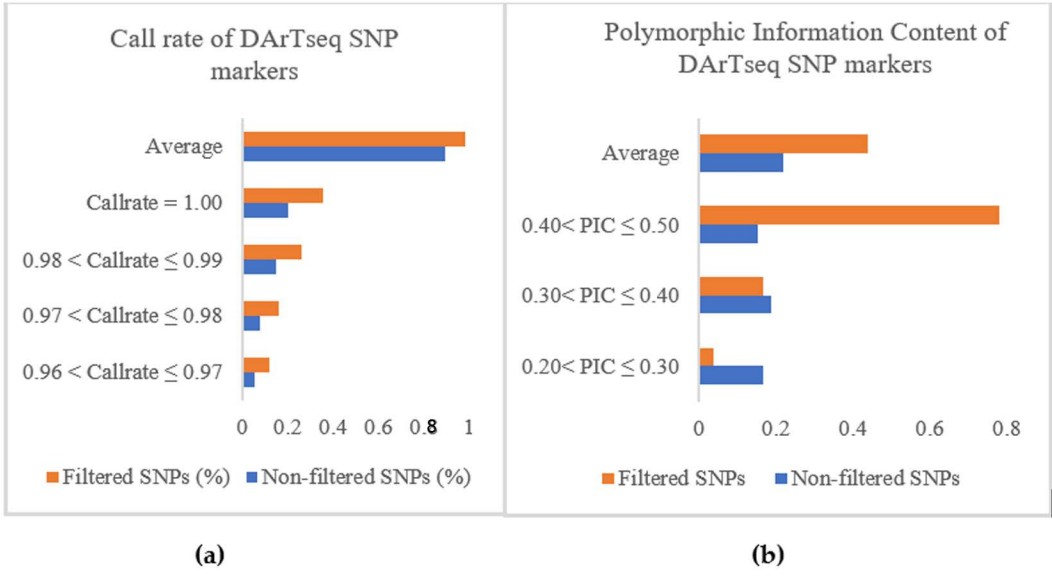

**Fig 1. Call rate (a) and Polymorphism Information Content (PIC) (b) of 8087 SNP markers.**

to 0.5. The average PIC improved from 0.22 before quality control to 0.44 after quality control. After the call rate and PIC were determined, the quality was also checked to determine the distribution of loci across the genome (Fig 2). All 8,087 DArTseq loci were mapped on the first 28 chromosomes.

The results revealed that 42.82% (3,463 loci) of the loci were located on only the 3 first chromosomes, 41.87% of which were located on chromosome 1. This chromosome hosts only 17.93% of all the loci (1,450), followed by chromosome 2 (1129 loci) and chromosome 3 (884 loci). The lowest number of loci was observed on chromosome 16, which hosts 5 loci, followed by chromosome 25, with 12 loci. Alternative alleles of DArTseq SNPs were counted across the chicken genome via the total dataset of 8087 markers (Fig 3). DArTseq SNP alternative allele counts ranged from 12 to 155 alleles per locus, with an average of 57.72 alleles and a median of 55 alleles.

A comparison of the number of loci between populations is shown in Fig 4. On the one hand, the Rwamagana population hosted almost all the loci (8,086 DArTseq SNPs), followed by the Bugesera population, which hosted 8,080 DArTseq SNPs.

The SASSO population had a genome with a lower number of loci (7,281 DArTseq SNPs). This was confirmed by the statistical parameters. The averages were 31.45, 19.35, and 7.71 alleles per locus for the Rwamagana, Bugesera, and SASSO populations, respectively, with ranges of 1–93, 1–56, respectively. The respective medians were 30, 18, and 7 alleles.

### Genetic variability and genetic relationship between chicken populations

The genetic distance between individuals within populations was estimated by genetic distance indices (S3 Table), while Nei's genetic distance matrix and Fst (Table 1) were used to analyze the genetic dissimilarities among chicken populations. The observed heterozygosity (Ho), expected heterozygosity (He), and fixation index (F) were used to analyze genetic relationships within populations, and the individual chickens were studied. The average distance among all individuals within the three chicken populations was $0.344\pm0.026$, ranging from 0.145 to 0.431. The least genetic distance was observed in the SASSO population ($0.259\pm0.019$), with a range of 0.145 to 0.285, whereas the Bugesera population presented greater genetic distance ($0.337\pm0.019$) ranging from 0.238 to 0.381. The genetic distance between individuals

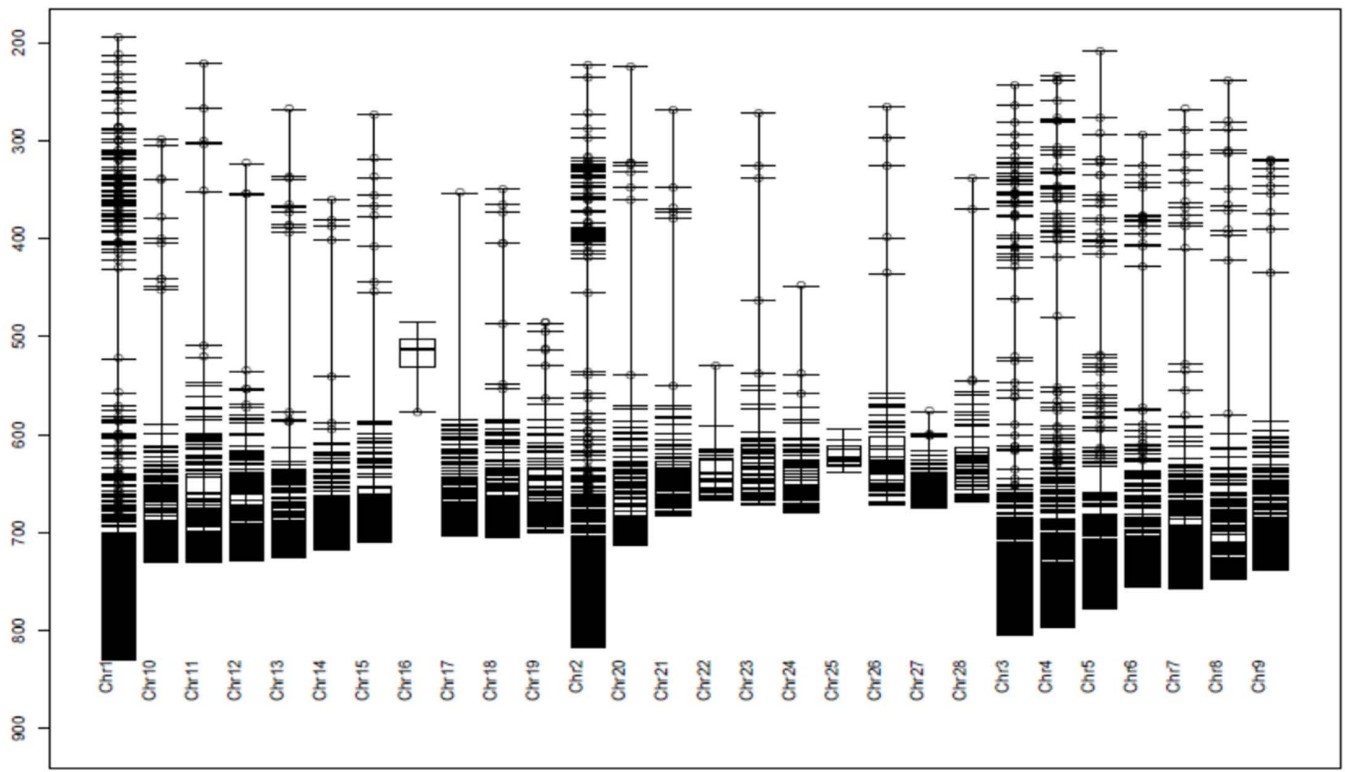

**Fig 2. Distribution of loci on 28 chromosomes (chr: chromosome).**

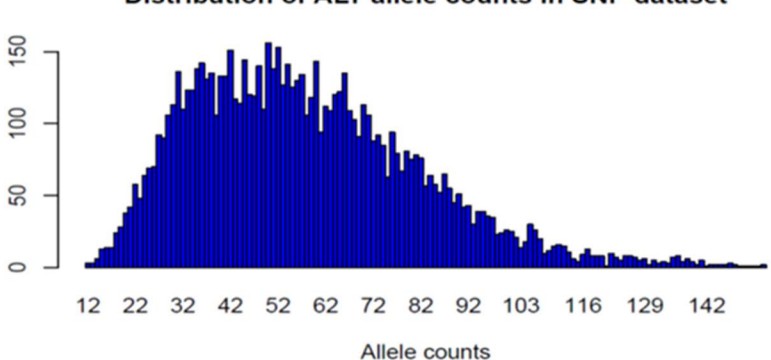

**Fig 3. Distribution of SNP alleles and loci in the total dataset.**

within the Rwamagana population ranged from 0.165 to 0.405, with an average of 0.334±0.022, but was closer to that of the Bugesera population. Indigenous chickens presented high genetic dissimilarity between individuals (0.338±0.020), ranging from 0.165 to 0.427, and among individuals of the SASSO population. Table 1 shows the genetic similarity among the Bugesera and Rwamagana populations, with a genetic distance of 0.009, similar to Fst, indicating that the two populations were genetically similar. The genetic dissimilarities were moderately low between the SASSO population and the

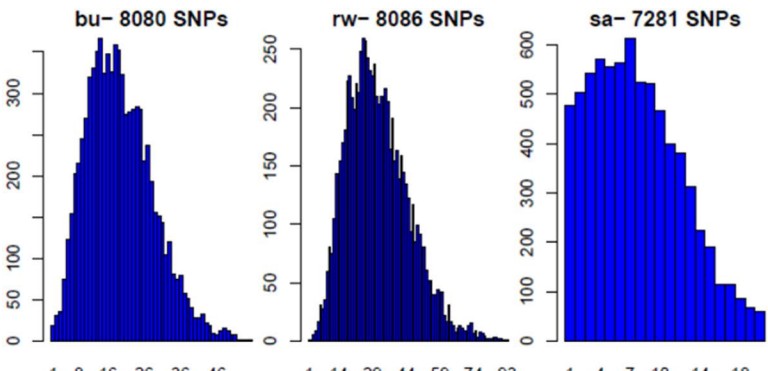

**Fig 4. Distribution of alternate SNP allele counts in each population (bu:bugesera population; rw: Rwamagana population; sa: SASSO population).**

**Table 1. Genetic distance among (Nei's 1972/Fst) and within (heterozygosity) populations, and the fixation index (F) observed via DArTseq SNP markers.**

| Nei's Genetic distance/Fst | | | | N | Ho | He | F | PL% |
|---|---|---|---|---|---|---|---|---|
| | Rwamagana | SASSO | | | | | | |
| Bugesera | 0.009/0.009 | 0.020/0.115 | Mean | 29 | 0.337 | 0.402 | 0.173 | 100.0 |
| | | | SE | | 0.001 | 0.001 | 0.003 | |
| Rwamagana | | 0.018/0.091 | Mean | 47 | 0.339 | 0.405 | 0.177 | 100.0 |
| | | | SE | | 0.001 | 0.001 | 0.002 | |
| SASSO | | | Mean | 10 | 0.316 | 0.388 | 0.181 | 98.9 |
| | | | SE | | 0.002 | 0.001 | 0.004 | |
| Total | | | Mean | 28.7 | 0.330 | 0.398 | 0.177 | 99.6 |
| | | | SE | 0.097 | 0.001 | 0.001 | 0.002 | 0.4 |

$F_{st}$ = Fixation index between populations; He = Expected heterozygosity; Ho = observed heterozygosity; F = Fixation index within populations; N = simple size; PL = Polymorphism level; SE = standard error.

other two populations, with genetic distances of 0.020 and 0.018 between Bugesera and SASSO and between Rwamagana and SASSO, respectively. This was confirmed by the difference in Fst between the SASSO population and the other two populations. It was 0.115 and 0.091 between the SASSO and Bugesera population and between the SASSO and Rwamagana populations, respectively. However, all the populations presented slightly greater genetic diversity, as explained by the expected heterozygosity values of 0.402, 0.405, and 0.388 for the Bugesera, Rwamagana, and SASSO populations, respectively.

In general, all the populations presented high genetic variability, as confirmed by the general expected heterozygosity of 0.398. In all the cases, the expected heterozygosity was greater than the observed heterozygosity. The inbreeding coefficient was low in all populations (not more than 0.18). All the DArTseq SNP markers were polymorphic (100%) in the Bugesera and Rwamagana populations, whereas 99.6% were polymorphic in the whole chicken dataset. The results of genetic dissimilarity revealed that the Bugesera and Rwamagana populations are genetically closer, but the two populations are genetically distant from the SASSO population. This was similarly visualized in the dendrogram of the chicken populations, which were divided into two distinct groups (Fig 5): one group was composed of SASSO (Sa) chickens, and

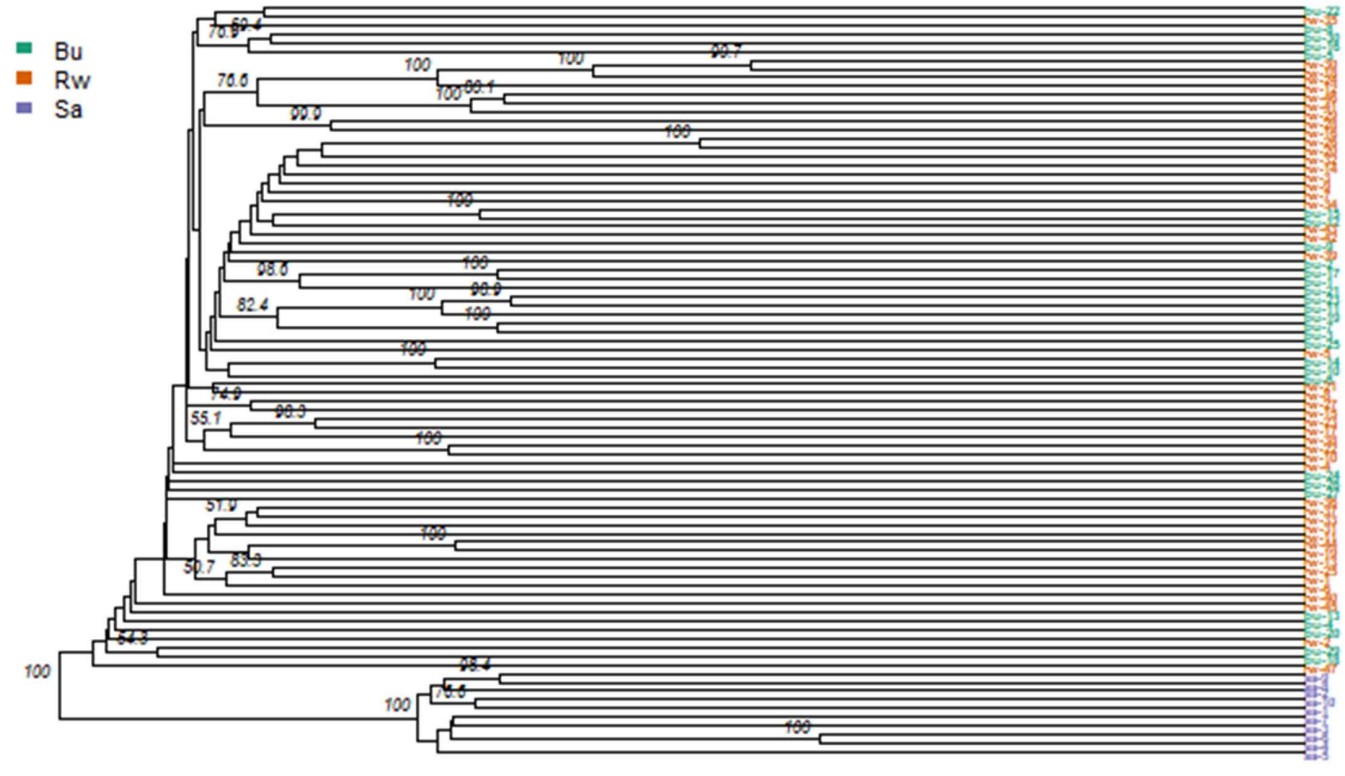

**Fig 5. Dendrogram for populations based on SNPs Markers (Bu: Bugesera population; Rw: Rwamagana population; Sa: SASSO).**

the other group was composed of Bugesera (Bu) and Rwamagana (Rw) chickens. This clearly revealed that the indigenous chickens were genetically different from the SASSO chickens.

## Analysis of the population structure of chicken samples

After the analysis of genetic diversity and the relationships among the chicken populations, the population structure was also analyzed to determine the extent of segregation among the chicken samples via principal component analysis (PCA) and the Bayesian clustering model of STRUCTURE software. Principal component analysis (PCA) performed via the dartR package revealed that the chicken samples were grouped into two clusters (Fig 6). The first cluster grouped chickens from the SASSO breed, whereas the second cluster comprised chickens from Bugesera and Rwamagana.

The PCA revealed that 12 chickens from Rwamagana were admixed with SASSO. The Bayesian clustering model of STRUCTURE software revealed the, for both subpopulations, the SASSO chickens were far separated from the indigenous chickens from K2 to K6 and formed a distinct group (Fig 7). At K6, Group 5 was the largest, with 32 chickens, followed by Group 4, with 23 chickens. Group 5 included 17 chickens from Bugesera and 17 chickens from Rwamagana, whereas group 4 included 19 chickens from Rwamagana and 4 chickens only from Bugesera. Similar to K6, at K5, many chickens were assigned to group 6, with 30 birds comprising 16 chickens from Bugesera and 14 chickens from Rwamagana.

Group 5 included 24 chickens, including 4 birds from Bugesera and 20 from Rwamagana. At K4, group 3, with 46 indigenous chickens, was the largest, followed by group 2, with 19 indigenous chickens. Group 3 comprised 15 birds from Bugesera and 31 birds from Rwamagana, whereas group 2 comprised 13 birds from Bugesera and 6 birds from Rwamagana.

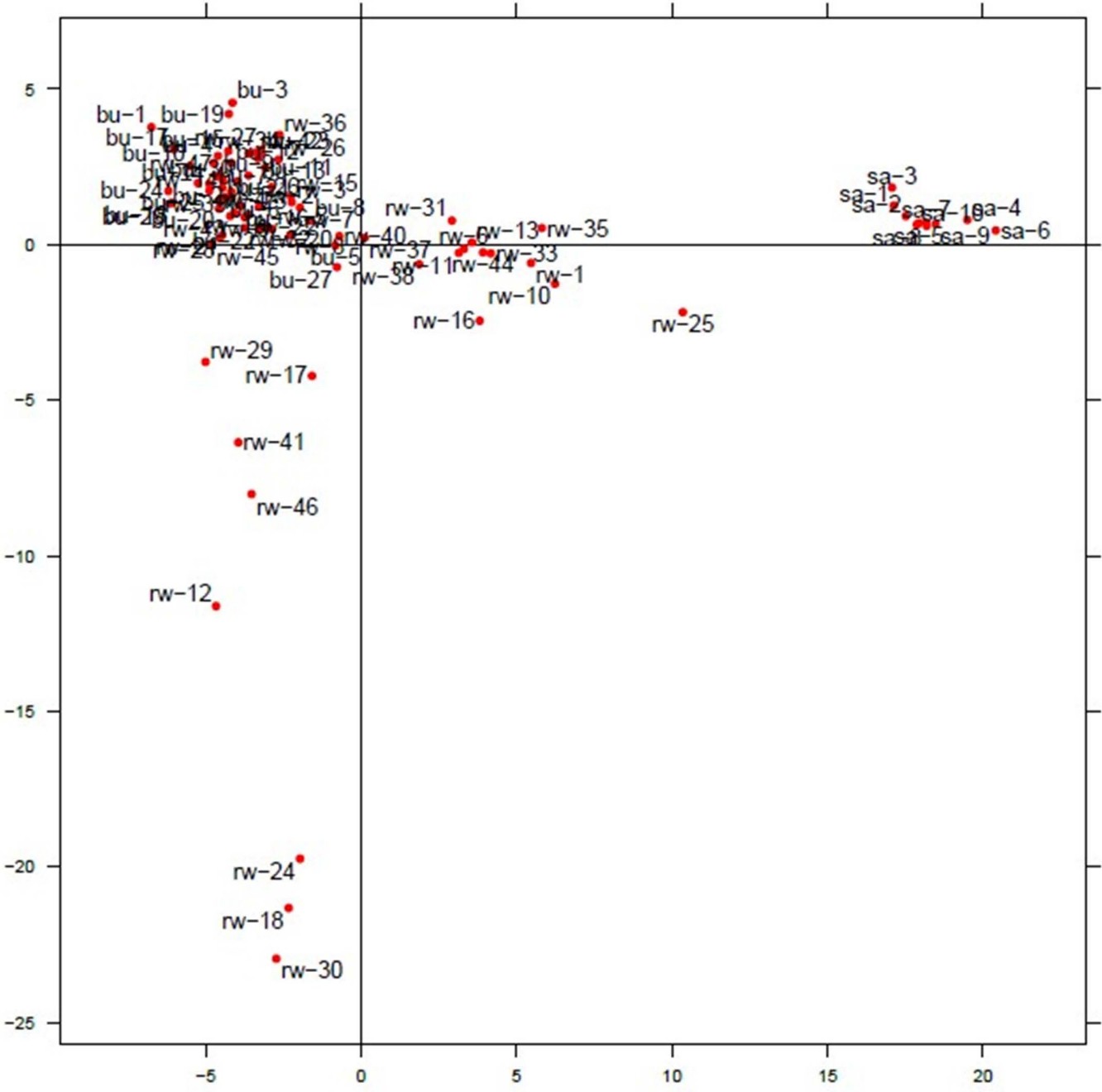

**Fig 6. PCA of chickens from Bugesera, Rwamagana and SASSO populations.**

At K3, most indigenous chickens were assigned to group 3, with 63 birds comprising 28 chickens from Bugesera and 35 birds from Rwamagana. Referring to ΔK=509.5, the logarithm probability, generated by STRUCTURE Harvester software, was used to determine the real number of groups. This parameter revealed that the chicken samples were grouped into two different clusters. Cluster 1 included SASSO chickens, whereas cluster 2 included chickens from Bugesera and Rwamagana.

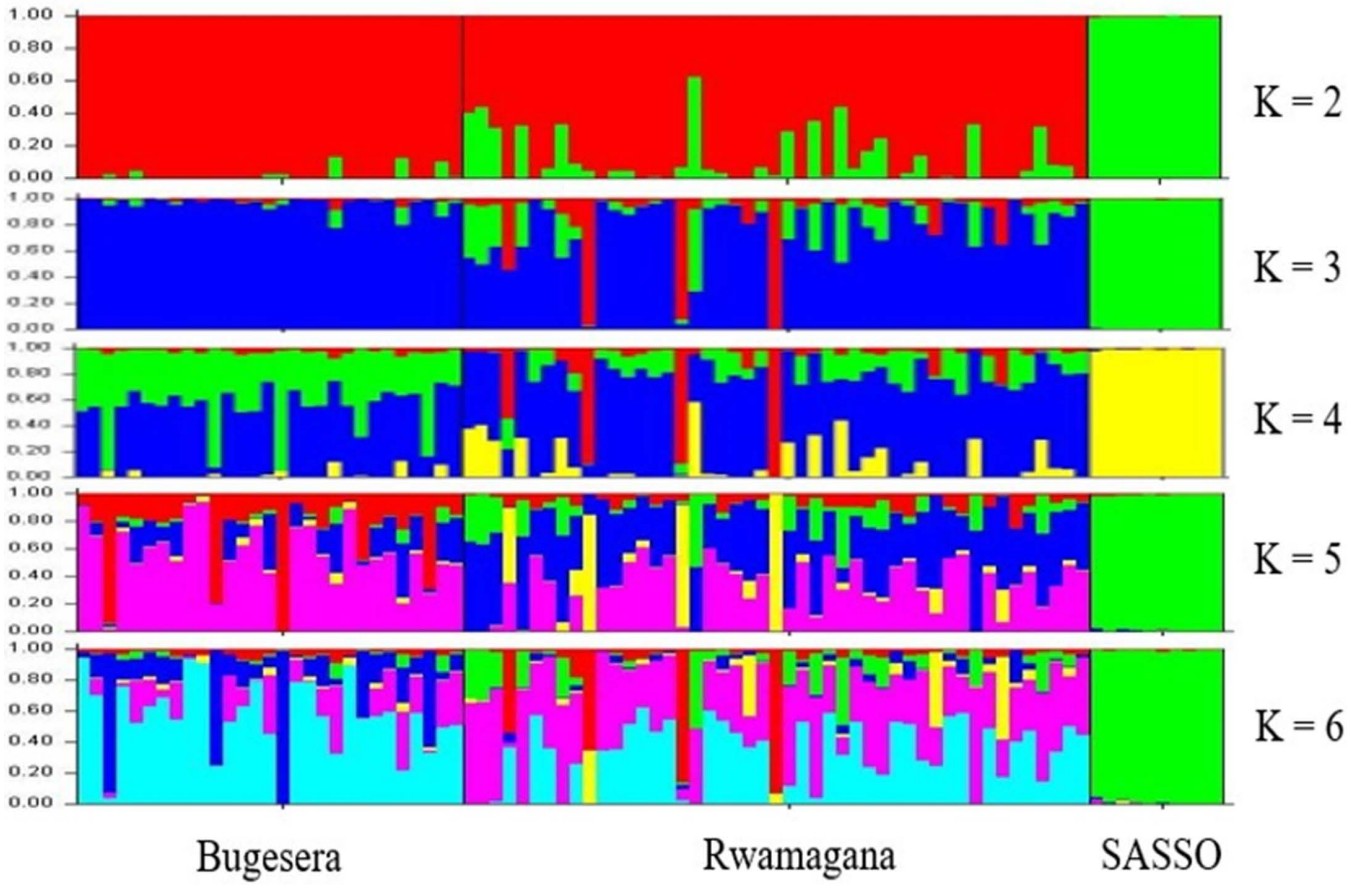

**Fig 7. The number of clusters (K) and population structure of chicken genotypes.**

However, the respective memberships of the two clusters were 19.3% and 80.7% for cluster 1 and cluster 2, respectively (Table 2). This means that cluster 1 grouped 17 chicken genotypes, whereas cluster 2 grouped 69 chicken genotypes.

The results revealed that all SASSO chickens were assigned to cluster 1, indicating that 7 chickens assigned to this cluster were from other chicken populations. This explains the presence of admixture in cluster 2. All 7 chickens admixed with SASSO were from the Rwamagana population, and were detected using an ancestry percentage of ≥ 25%. However, cluster 2 included 40 chicken genotypes from the Rwamagana population and all 29 genotypes from Bugesera. This admixture explains the close relationship observed between the Rwamagana population and SASSO chickens, confirming what was observed with Nei's genetic distance and PCA above. The two clusters presented high genetic variability explained by the expected heterozygosity (He). Cluster 2 had greater genetic variability (He = 0.404) than did cluster 1 (He = 0.361). The inbreeding coefficient represented by Fst was lower in cluster 2 (Fst = 0.012) than in cluster 1 (Fst = 0.218). This confirms the results from genetic diversity generated by GenAlEx software described above. The results from the analysis of molecular variance (AMOVA) corroborate the results from genetic relationships and population structure. The variance was significant between chicken populations (Table 3), with p ≤ 0.01. The variance between populations was very low (4.32% of the total variance); however, high variance was observed among the chicken individuals (80.30%).

**Table 2. Characteristics of clusters of 86 chickens.**

| Net nucleotide distance | | Membership probability | He | $F_{st}$ |
|---|---|---|---|---|
| | **Cluster 2** | | | |
| **Cluster 1** | 0.0440 | 0.193 | 0.361 | 0.218 |
| **Cluster 2** | | 0.807 | 0.404 | 0.012 |

**Table 3. Analysis of molecular variance (AMOVA) via SNP markers between and within populations of 86 chicken genotypes.**

| Source of variation | Df | SS | MS | Component of variance | % of variance | P values |
|---|---|---|---|---|---|---|
| **Among populations** | 2 | 10962.03 | 5481.02 | 73.01 | 4.32 | 0.01 |
| **Among samples within populations** | 83 | 155727.66 | 1876.24 | 259.84 | 15.38 | 0.01 |
| **Within individuals** | 86 | 116664.50 | 1356.56 | 1356.56 | 80.30 | 0.01 |
| **Total** | 171 | 283354.19 | 1657.04 | 1689.41 | 100.00 | |

Df = degree of freedom; SS = sum of squares; MS = mean of squares.

## Discussions

### DArTseq SNP marker quality and distribution in the chicken genome

The results revealed that quality control improved the call rate (Fig 1a) and the polymorphic information content, especially the PIC, between 0.4 and 0.5 (Fig 1b) of the markers used. The call rate in our study was slightly similar to those reported in horses (0.97) and sheep (0.97), and slightly higher than the call rate reported by Gurgul et al. [26] for bovines (0.95). Notably, a call rate > 80%, minor allele frequency (MAF) > 0.01, and HWE with p > 0.001 were the criteria for SNP quality control for the species used in the studies above. This study revealed that almost all the DArTseq SNPs used were able to assign all the genotypes. The results from this study elucidated the resemblance in terms of locus quality control. Polymorphism is the parameter used to characterize the quality of genetic markers and confers the ability to be selected for genomic studies. According to Serrote et al. [27], a genetic marker is not effective if it is unable to detect the genetic dissimilarities present among a given number of individuals. The polymorphism of a marker can be qualitatively or quantitatively characterized. Qualitatively, a marker is said to be polymorphic when two alleles are present, and the frequency of the most frequent allele increases to 99%, as mentioned by Shete et al. [28], whereas quantitatively, the heterozygosity in the study of Nei and Roychoudhury [29] and the polymorphic information content (PIC) in the study of Botstein et al. [30] are used to measure the marker's polymorphism. The number of DArTseq SNP polymorphic loci (95%) was greater than the 82.3% of polymorphic microsatellite loci observed in the study by Habimana et al. [6]. The difference might be due either to the marker's features or to the PIC estimation method. The microsatellites are codominant markers, whereas the SNPs are biallelic markers with presence or absence (1,0) scores. The PIC estimation for codominant markers is based on the allele frequency, with PIC values ranging from 0 (monomorphic markers) to 1 (highly informative) [31–33], where PIC values less than 0.25 provide less information [30]. For codominant but biallelic markers such as SNPs [34], the PIC values range from 0 (monomorphic markers) to 0.5 (markers available in 50% of individuals and absent in the remaining 50%) and are estimated via heterozygosity in a similar way [31,35] for dominant markers, 0.5 is the maximum value [32]. Ranges of PIC values of 0–0.10, 0.10–0.25, 0.25–0.40, and 0.40–0.50 are described as low, medium, high, and very high, respectively [31]. These relative ranges offer high-quality DArTseq SNPs used in this study, which are highly polymorphic, and have been recommended for genetic studies in indigenous chickens. The polymorphic DArTseq SNP loci produced in this study were distributed across 28 chromosomes from 5 loci (chromosome 16) to 1450 loci (chromosome 1). A total of 42.87% (3,463 loci) of all the loci were located on the first 3 chromosomes. This might explain the different sizes of the 38 pairs of autosomal chromosomes in the chicken genome. The differences in chromosome sizes can be confirmed by the

information provided in Fig 2. The first 5 chromosomes are longer than the other chromosomes, and they host more than half of the polymorphic DArTseq SNP loci used in this study. As shown in Fig 2, the sizes of the chromosomes decrease progressively from chromosome 1. Chicken chromosomes are conventionally classified into three groups: a) those from chromosomes 1–5 are described as macrochromosomes as they are larger than 40 Mbp, b) those from chromosomes 6–12 are described as intermediate chromosomes, and c) the remaining chromosomes are described as micro-chromosomes [36,37]. Although some micro-chromosomes host a relatively large number of loci, the polymorphic DArT-seq SNP markers located on the first five macrochromosomes, particularly on chromosome 1, could be sufficient for investigating genetic diversity, population structure, and other genomic studies in IC populations. The results of this study revealed that there was no difference in the distribution of DArTseq SNPs in the chicken genome between populations. Similar results showing a high number of SNP markers on chromosome 1 were observed in horses and sheep [26], with 5.4% and 9.2% polymorphic markers, respectively. The results of this study disagree with what was observed in the bovine study mentioned by Gurgul et al. [26] and that, with the exception of chromosome X, autosome 4 was the chromosome that hosted the greater number of markers, representing 4.6% of the polymorphic SNPs. In all cases, the results show that SNP coverage is more important on the first chromosomes in chickens than it is in these species, in which the variation in the distribution of markers on chromosomes can be low. In terms of the alternate allele distribution across all the loci, in SASSO, 534 loci presented no alternate alleles. This type of chicken presented a greater number of loci (475 loci), with 1 alternate allele. The results in Table 1 show the differences observed between populations. In the Bugesera and Rwamagana populations, which presented a low number of loci with 0 or 1 alternate allele, all the loci (100%) were polymorphic, whereas, in the SASSO population, only 98.9% were polymorphic (Table 1). The results for polymorphic loci in other animal species were similar to those for chickens in the current study. It ranges from 0.97 to 0.99 in Australian goat breeds, and from 0.90 to 0.95 in Canadian goat breeds [38]. The implication of the presence of a high number of loci presenting low alternate alleles is the decrease in marker polymorphism, which might indicate the level of allele fixation at that locus in the population. Local breeds tend to have a greater number of polymorphic loci than exotic breeds do. The results from the current work confirmed those observed in the work of Mollah et al. [39] via random amplified polymorphic DNA (RAPD) markers. The quality of the DArTseq SNP markers in terms of the genotype calling rate, high polymorphism, and high number of alternate alleles across the genome make them suitable for chicken genetic studies, including genetic diversity and population structure.

### Genetic variability and genetic relationships between chicken populations

In designing or updating breeding and conservation programs, genetic diversity characterization within population is crucial [38]. The genetic variability and relationships between and within populations were analyzed with genetic distance indices (S3 Table), Nei's genetic distance matrix, heterozygosity (Table 1), and dendrograms (Fig 5). The genetic distance index between all 86 individual chickens was $0.344 \pm 0.026$, indicating that they were genetically distant. Bugesera and Rwamagana chickens are genetically closer and moderately distant from SASSO chickens. The genetic diversity represented by expected heterozygosity (He) was slightly lower in SASSO and quite similar in Bugesera and Rwamagana ICs. In all the cases, in the present study, the expected heterozygosity was greater than the observed heterozygosity. This explains the potential fixation of some alleles in the populations, which could result in an increase in inbreeding. This situation could be because the mating systems in the study area might not be random, as most households use birds from their own flocks or from close neighbors. In addition, mating occurs between parents and progeny or between siblings, half-sibs, cousins, etc., which could result in deviation from Hard-Weinberg equilibrium in these populations. The loci were highly polymorphic in the populations, which explains their segregation. Contrasting results were observed in Chinese local chickens, in which He was either lower than or equal to Ho [40,41]. Therefore, the discrepancy in genetic distances observed within populations was due to co-ancestry mating, and the level of inbreeding. However, the absence of intensive selection within the native chicken populations, unlike the SASSO population, makes them an outbred population,

which is the main reason for the variations between the two types of chickens. This can be confirmed by the high genetic distance observed, which points to the presence of high heterogeneity in native chickens. The low genetic dissimilarity and low genetic variability observed in SASSO chickens could be due to intensive selection of productive traits and inbreeding resulting in the fixation of alleles. This observation corroborates the findings of [42,43], that genetic variability in commercial breeds was reduced to 50%. Similarly, low genetic diversity in local chickens was observed in Chinese layer lines [40]. This confirmed what was observed in goat populations in Australia [41], where the genetic diversity and genetic distance in Rangeland goats were greater than those in the Toggenburg breed, which experienced artificial selection. However, the genetic distances observed in this study were slightly similar to those reported in Rangeland goats in Australia (0.323) and slightly greatly than those reported in other Australian and Canadian goats, ranging from 0.263 to 0.307 [38]. This study revealed that IC populations are more closely related, and distant from SASSO chickens (improved breeds), as confirmed by the dendrogram (Fig 5). The absence of genetic distance between IC populations could be explained by the fact that these populations are very close neighbors, as they have a common border; thus, the share of genetic materials implies that the two populations have a common ancestor. These findings corroborate the conclusion of Wright [44], who stated that the geographical closeness between populations influences the genetic relatedness between them. Another possibility is the large chicken population in this region [6], which helps to conserve their alleles and prevent mutations. Our study confirmed several earlier findings on the small genetic distances and close genetic relationships among native chicken populations in Rwanda [6], Iran [45], and India [43].

## Analysis of the population structure of chicken samples

The analysis of chicken population structure was performed via PCA (Fig 6), the STRUCTURE program (Fig 7), and AMOVA (Table 3). All the chicken genotypes were correctly clustered on the basis of the origin of the IC population, and were separated from the exotic breed used (SASSO). This means that all the ICs have the same genetic background and similar management, which are different from those of the SASSO breed. The Bugesera and Rwamagana populations were grouped into one group, and the SASSO chickens were grouped into another group. The individuals clustered together indicate that they share the same genetic background or have a common ancestor. As mentioned earlier, the Bugesera and Rwamagana populations are closer to each other. However, although they are assigned to the same group, they are different geographically, indicating that there is genetic differentiation between them. A similar classification of IC based on geographical location was observed in Rwanda [6] and in Italian local chickens [46]. The same classification was also reported in other species in Italy by Ciani et al. [47] in sheep breeds, in goat breeds [48], and in cattle by Mastrangelo et al. [49] and Gurgul et al. [26]. However, in our study, the chicken genotypes were grouped on the basis of geographical classification for indigenous chicken individuals and genetic classification for all chicken individuals. A study by Zhang et al. [40] reported similar results, with the classification distinguishing exotic chicken breeds from Chinese IC breeds exhibiting consistency in their genetic structure, as observed in this study. Twelve chickens and 7 chickens identified via PCA and STRUCTURE, respectively, were admixed with SASSO chickens, and were all from the Rwamagana population. This provides information on where attention should be placed when planning breeding and conservation strategies for Bugesera and Rwamagana populations. Before any planning, the initial analysis of population structure, including all genetic resources present, could be necessary, particularly in the Rwamagana population and in all Rwandan chicken populations in general. This will help to identify individuals with pure genetic profiles that will serve as a starting point for breeding and gene conservation programs. The DArTseq SNP markers used in this study presented a high number of variants in IC, and are suitable for further genomic studies in Rwandan IC. Assuming that these markers are linked to economically important QTLs, these markers are certainly able to identify them, allowing the setting of breeding objectives that can serve as targets of a successful genomic selection approach in IC. This assumption can be explained by the studies of González et I. [50] and Hu [51], who confirmed the association between allele counts and phenotypes for a specific trait of economic importance. The population structure and classification were supported by the AMOVA results,

which revealed that the distinction between populations was significant (P<0.01), indicating a difference between IC and SASSO chickens. A high proportion of variance was observed between genotypes, represented by 80.3%. The value observed in this study was >15%, reported as an indicator of significant dissimilarity between samples within populations [52]. A comparable situation was observed in a previous study by Habimana et al. [6], who reported a variance of 92%. Similar observations were reported in other studies on IC population structure, such as [53] in Italy, [54] in Ethiopia and Cameroon, and [55] in Tanzania.

## Conclusions

The chicken genotypes studied via the use of DArTseq SNP markers varied genetically and geographically. The IC populations presented high genetic variability but were genetically distant from the SASSO chickens. The SASSO chicken population was revealed to be genetically related to indigenous chickens from Rwamagana, resulting from gene flow through uncontrolled cross-breeding between SASSO chickens and IC. In addition, as SASSO chickens exhibit improved adaptability to harsh environments like IC, studies on their crossbreeding with IC could be encouraged to investigate the performance of their progeny under free-ranging management conditions. Thus, due to their genome coverage and high polymorphism, DArTseq SNP markers are suitable for genomic studies in chickens, and the genetic variability observed in Rwandan IC could be useful in planning genetic improvement and conservation programs. Therefore, functional validation of the identified DArTse SNP markers remains a priority for future research.

## Supporting information

**S1 Checklist. Inclusivity in global research.**
(PDF)

**S1 Table. DArTseq SNPs generated used in this study.**
(CSV)

**S2 Table. Metadata.**
(XLSX)

**S3 Table. Genetic distances within and between chicken populations.**
(XLSX)

**S1 File. Poultry Sector Analysis of Rwanda.**
(PDF)

**S2 File. Rwanda Livestock Master Plan.**
(PDF)

**S3 File. Integrated Household conditions Survey.**
(PDF)

**S4 File. Agricultural Household survey 2020.**
(PDF)

## Acknowledgments

The authors acknowledge and address special thanks to the SEAD (Strengthening Education for Agricultural Development) project for their support. The gratitude is also expressed to the SEQART Africa team for their assistance in laboratory work.

## Author contributions

**Conceptualization:** Valentin Mujyambere.

**Data curation:** Valentin Mujyambere.

**Formal analysis:** Valentin Mujyambere.

**Investigation:** Valentin Mujyambere.

**Methodology:** Valentin Mujyambere.

**Project administration:** Valentin Mujyambere.

**Resources:** Valentin Mujyambere.

**Software:** Valentin Mujyambere.

**Validation:** Valentin Mujyambere, Kwaku Adomako, Martin Ntawubizi, Laetitia Nyinawamwiza, Judith Uwihirwe, Alexander Wireko Kena.

**Visualization:** Valentin Mujyambere.

**Writing – original draft:** Valentin Mujyambere.

**Writing – review & editing:** Kwaku Adomako, Martin Ntawubizi, Laetitia Nyinawamwiza, Judith Uwihirwe, Alexander Wireko Kena.

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
