## [Decision Letter · Decision Letter 0]

21 Apr 2025

Dear Dr. Mujyembere,

Thank you for submitting your manuscript to PLOS ONE. After careful consideration, we feel that it has merit but does not fully meet PLOS ONE’s publication criteria as it currently stands. Therefore, we invite you to submit a revised version of the manuscript that addresses the points raised during the review process.

Both reviewers are happy with the aim and structure of the manuscript. However, they pointed out some minor amendments needed to accept the manuscript.

We look forward to receiving your revised manuscript.

Kind regards,

Gyaneshwer Chaubey

Academic Editor

PLOS ONE

Journal Requirements:

Reviewers' comments:

Reviewer's Responses to Questions

**Comments to the Author**

1. Is the manuscript technically sound, and do the data support the conclusions?

Reviewer #1: Yes

Reviewer #2: Partly

2. Has the statistical analysis been performed appropriately and rigorously?

Reviewer #1: Yes

Reviewer #2: Yes

3. Have the authors made all data underlying the findings in their manuscript fully available?

Reviewer #1: Yes

Reviewer #2: Yes

4. Is the manuscript presented in an intelligible fashion and written in standard English?

Reviewer #1: Yes

Reviewer #2: Yes

Reviewer #1: Review comments uploaded. .

Reviewer #2: The title of the manuscript, Genetic diversity and relationship of Bugesera and Rwamagana indigenous chicken populations with SASSO chickens using DArTseq SNPs, is good and the experimentations done to prove the aim of the work has been satisfactory, but one major thing which is not revealed anywhere, the distribution of loci at any particular chromosome is still ambigous, author must explain as written in Figure 2. Distribution of loci on 28 chromosomes and its legend.

The other concern is that in result section there is reported 45677 alleles in the present work, however, very few were grouped and represented into figure 4, need to explain about remaining alleles.

**Do you want your identity to be public for this peer review?** For information about this choice, including consent withdrawal, please see our Privacy Policy

Reviewer #1: No

Reviewer #2: **Yes: ** Dr. Yashvant Patel

---

## [Author Response · Author response to Decision Letter 1]

6 Jul 2025

Rebuttal letter to reviewers’ comments

PONE-D-24-58510 Genetic diversity and relationship of Bugesera and Rwamagana indigenous chicken populations with SASSO chickens using DArTseq SNPs

Dear Editor,

Thank you for the opportunity to revise and submit our manuscript “PONE-D-24-58510 Genetic diversity and relationship of Bugesera and Rwamagana indigenous chicken populations with SASSO chickens using DArTseq SNPs”. We sincerely appreciate the time and effort that you and reviewers have devoted to evaluate our work. We have carefully considered all the comments and suggestions provided, and we have revised the manuscript accordingly. In this letter, we provided a detailed response to each comment, indicating how we have addressed them in the revised version. We hope the revisions have strengthened the manuscript and addressed the concerns raised.

Editor

Comment 1: Please ensure that your manuscript meets PLOS ONE's style requirements, including those for file naming.

Answer: Authors tried to meet all PLOS ONE’s style requirements, following what is stated in templates. Those include

- Title

- Headings: throughout the manuscript

- Figures: throughout the manuscript

- References: since some were reports not meeting all requirements for referencing, they were cited and mentioned as supplementary files especially in the section of introduction

P 3 – P3: L 48 – L 71

- The list of supplementary materials and files was updated:

P 30: L 780 – L 787

Comment 2: Please include a complete copy of PLOS’ questionnaire on inclusivity in global research in your revised manuscript. Our policy for research in this area aims to improve transparency in the reporting of research performed outside of researchers’ own country or community. The policy applies to researchers who have travelled to a different country to conduct research, research with Indigenous populations or their lands, and research on cultural artefacts. The questionnaire can also be requested at the journal’s discretion for any other submissions, even if these conditions are not met. Please find more information on the policy and a link to download a blank copy of the questionnaire here: https://journals.plos.org/plosone/s/best-practices-in-research-reporting. Please upload a completed version of your questionnaire as Supporting Information when you resubmit your manuscript

Answer: Thank you! The comment was considered.

- The inclusivity in global research was included in the manuscript, the section of Materials and Methods.

P 7: L 199 – L 201

It was also added as a supplementary checklist (S1 Checklist)

P 30: L 780

- The Informed Consent Statement was also included in the manuscript, the section of Materials and Methods.

P 8: L 192 – L 196

Comment 3: Please review your reference list to ensure that it is complete and correct. If you have cited papers that have been retracted, please include the rationale for doing so in the manuscript text, or remove these references and replace them with relevant current references. Any changes to the reference list should be mentioned in the rebuttal letter that accompanies your revised manuscript. If you need to cite a retracted article, indicate the article’s retracted status in the References list and also include a citation and full reference for the retraction notice

Answer: The reference list was reviewed and some references were removed and replaced by the supplementary files.

P 3 – P3: L 48 – L 71

These supplementary files were added to the list of supplementary information:

P 30: L 784 – L 787

Reviewer # 1

Comment 1: Since at least five studies have been published after the LMP's implementation, it would be helpful to include findings from these studies. Specifically, how do the current statistics compare to previous levels of inbreeding? Have these studies demonstrated a measurable improvement in genetic variability among the birds? Including this information would provide a more comprehensive understanding of the program’s effectiveness.

Answer: Some findings from studies were included in the introduction to link them with the LMP.

P 4 – P5: L 88 – L 113

Comment 2: A question for clarity: Why were the indigenous chickens (IC) sampled at ages 8 to 10 months, while only female SASSO chickens were sampled precisely at 9 months of age?

P 7: L 163 – L 168

Answer: Thank you for the question. The difference in age of sampling between IC and SASSO females was based on two main factors:

- Birds’ maturity: the aim was to use chickens already in reproduction age. Since IC are late and heterogeneous in entering mature age, most of them reach the body size and start laying between 7 to 10 months of age. However, SASSO are improved dual-purpose breeds (meat and eggs) and are fast growers, they can reach maturity around 5 – 6 months.

- Environmental factors: variation in maturity depends on the interaction between genetics and environment. These IC were sampled from different households with different management practices, which may cause the variability in attaining the maturity. Then, only SASSO females were considered in this study because they were sampled from one intensive egg production farm. They were at the age of 9 months. Since no SASSO males were available, the screening took into account the loci on the sex chromosomes to avoid bias in findings.

Comment 3: Towards the end of the discussion on genetic variability and genetic relationships between chicken populations, you briefly mention how the absence of intensive selection within native chicken populations leads to inbreeding, contributing to genetic variation between the two types of chickens. However, I found this part somewhat unclear, especially in relation to your earlier point about mating systems in the study area potentially not being random. It might be helpful to elaborate further on this connection as I was unsure about the exact argument being made and the direction you were leading us toward in this discussion.

Answer: You are right! The confusion is no longer there.

The “inbred” was replaced by the “outbred”.

P 21: L 518

Comment 4: I suggest mentioning the authors' names when referencing them in-text before including the numbered citation. Using only a number, such as (1), disrupts the readability of the sentence. For example, instead of writing:

"There was an increase in poultry production, reported in 2014 by (2),"

it would read more smoothly as:

"There was an increase in poultry production, reported in 2014 by Sharpio (2)."

Answer: The comment considered and citations corrected:

P 3: L 57

P 18: L 429

P 18: L 435

P 18: L 439

P 18: L 440

P 18: L 441

P 18: L 444

P 20: L 475

P 20: L 491

P 22: L 535

P 23: L 556

P 23: L 557

P 23: L 559

P 24: L 573

P 24: L 580

Comment 5: There was an inconsistency in the in-text citations, as some were numbered while others were not. I noticed this particularly in the Methods section.

Answer: Thank you! Corrected

P 6: L 143

P 11: L 254

Comment 6: Throughout the manuscript, the number of chicken genotypes (87 or 86) was inconsistently written as "eight-seven" instead of "eighty-seven" or "eighty-six." I recommend correcting this for clarity and consistency.

Answer: Thank you! Corrected:

P 2: L 28: “Eight-seven” was replaced by “Eighty-seven”.

P 7: L 149: “Eight-seven” was replaced by “Eighty-seven”.

Comment 7: In the Abstract, the authors introduced low He, assuming readers would immediately recognize the term. To avoid confusion or unnecessary assumptions, I suggest replacing it with "expected heterozygosity (He)" for clarity right from the start of the paper.

Answer: Thank you! Corrected

P 2: L 32

Comment 8: Do not start a sentence with an abbreviation; instead, write it out in full when beginning a sentence. Thank you! Considered

P 3: L 66

P 10: L 240 – L 241

Comment 9: Please ensure that page numbers are included in the manuscript.

Answer: Thank you! Page numbers and line numbers were included in the manuscript

Reviewer # 2

Comment 1: One major thing which is not revealed anywhere, the distribution of loci at any particular chromosome is still ambiguous, author must explain as written in Figure 2. Distribution of loci on 28 chromosomes and its legend.

Answer: Thank you for this concern!

However, the interpretation of the distribution of loci on chromosomes as written in Figure 2, was extended:

- Legend added (chr: chromosome) P 12: L 283

- In results P 12: L 280 – L 293

- In discussions: P 19 – P 20: L 455 – L 479

Comment 2: The other concern is that in result section there is reported 45677 alleles in the present work, however, very few were grouped and represented into figure 4, need to explain about remaining alleles

Answer: Thank you for this concern! You are right.

However, the quality control of markers was performed so that only informative and good markers remain for data analysis. This for the following reasons:

- To ensure data accuracy and reliability: low quality markers (with lots of missing data for example) can introduce noise which can lead to false association, biased diversity estimates, and unreliable population structure results

- To remove non-informative or problematic markers: all markers which are monomorphic (no variation), having a very low Minor allele Frequency, etc.

- To improve statistical power and validity: this avoids misleading genetic diversity or population structure analysis and improves the reproducibility and scientific credibility of results.

The criteria used in this study for quality control were described in the section of materials and methods:

P 10: L 224 – L 241

---

## [Decision Letter · Decision Letter 1]

14 Aug 2025

Genetic diversity and relationship of Bugesera and Rwamagana indigenous chicken populations with SASSO chickens using DArTseq SNPs

PONE-D-24-58510R1

Dear Dr. Mujyambere,

We’re pleased to inform you that your manuscript has been judged scientifically suitable for publication and will be formally accepted for publication once it meets all outstanding technical requirements.

Kind regards,

Gyaneshwer Chaubey, PhD

Academic Editor

PLOS ONE

Additional Editor Comments (optional):

Reviewers' comments:

Reviewer's Responses to Questions

**Comments to the Author**

Reviewer #1: All comments have been addressed

2. Is the manuscript technically sound, and do the data support the conclusions?

Reviewer #1: Yes

3. Has the statistical analysis been performed appropriately and rigorously?

Reviewer #1: Yes

4. Have the authors made all data underlying the findings in their manuscript fully available?

Reviewer #1: Yes

5. Is the manuscript presented in an intelligible fashion and written in standard English?

Reviewer #1: Yes

Reviewer #1: (No Response)

**Do you want your identity to be public for this peer review?** For information about this choice, including consent withdrawal, please see our Privacy Policy

Reviewer #1: No

---

## [Editor Report · Acceptance letter]

PONE-D-24-58510R1

PLOS ONE

Dear Dr. Mujyambere,

I'm pleased to inform you that your manuscript has been deemed suitable for publication in PLOS ONE. Congratulations! Your manuscript is now being handed over to our production team.

Kind regards,

on behalf of

Prof. Gyaneshwer Chaubey

Academic Editor

PLOS ONE